# Canary in the coliform mine: Exploring the industrial application limits of a microbial respiration alarm system

**Wendy Stone**[1]*, **Tobi M. Louw**[2], **Marthinus J. Booysen**[3], **Gideon M. Wolfaardt**[1,4]

**1** Water Institute and Department of Microbiology, Stellenbosch University, Stellenbosch, South Africa,
**2** Department of Process Engineering, Stellenbosch University, Stellenbosch, South Africa, **3** Department of
E&E Engineering, Stellenbosch University, Stellenbosch, South Africa, **4** Department of Chemistry and
Biology, Ryerson University, Toronto, Canada

* wstone@sun.ac.za

pone.0247910

Sciences, CHINA

**Data Availability Statement:** All relevant data are
publicly accessible via Kaggle (https://www.kaggle.
com/wendystone/a-microbial-respiration-alarm-
system).

## Abstract

Fundamental ecological principles of ecosystem-level respiration are extensively applied in
greenhouse gas and elemental cycle studies. A laboratory system termed CEMS (Carbon
Dioxide Evolution Measurement System), developed to explore microbial biofilm growth and
metabolic responses, was evaluated as an early-warning system for microbial disturbances
in industrial settings: in (a) potable water system contamination, and (b) bioreactor inhibition.
Respiration was detected as $CO_2$ production, rather than $O_2$ consumption, including aerobic
and anaerobic metabolism. Design, thresholds, and benefits of the remote $CO_2$ monitoring
technology were described. Headspace $CO_2$ correlated with contamination levels, as well
as chemical ($R^2 > 0.83$–$0.96$) and microbiological water quality indicators ($R^2 > 0.78$–$0.88$).
Detection thresholds were limiting factors in monitoring drinking water to national and inter-
national standards (0 CFU/100 mL fecal coliforms) in both open- (>1500 CFU/mL) and
closed-loop $CO_2$ measuring regimes (>100 CFU/100 mL). However, closed-loop detection
thresholds allow for the detection of significant contamination events, and monitoring less
stringent systems such as irrigation water (<100 CFU/mL). Whole-system respiration was
effectively harnessed as an early-warning system in bioreactor performance monitoring.
Models were used to deconvolute biological $CO_2$ fluctuations from chemical $CO_2$ dynamics,
to optimize this real-time, sustainable, low-waste technology, facilitating timeous responses
to biological disturbances in bioreactors.

## Introduction

Carbon dioxide production is a universal biological indicator of respiration, and is a parameter
indicative of life, often harnessed as an indicator of ecosystem health [1, 2]. Pursuing a similar
bird's-eye-view, Kroukamp and Wolfaardt [3] developed the Carbon Dioxide Evolution Mea-
surement System (CEMS), which harnesses microbial $CO_2$ production to study whole-biofilm
metabolic profiles. Unlike standard respirometry [4], $CO_2$ rather than $O_2$ is monitored as the
by-product of glycolysis and the Krebs cycle. This metabolic pathway is common to aerobic

**Funding:** GMW and WS were funded by the European Union's Horizon 2020 research and innovation program (https://ec.europa.eu/programmes/horizon2020/en), grant agreement No 689925. The study reflects only the authors' views. The EU is not responsible for any use that may be made of the information it contains. The funders had no role in study design, data collection and analysis, decision to publish, or preparation of manuscript.

**Competing interests:** The authors have declared that no competing interests exist.

and anaerobic metabolism, and the measurement of $CO_2$ fluctuation is also indicative of photosynthetic metabolism. The system traps biofilm-produced $CO_2$, passing it over an analyzer on $CO_2$-free sweeper gas, and logging the data. Online platforms allow for remote, real-time monitoring of disturbances. It is used primarily to study pure culture laboratory biofilms, and has generated information on the relationship between biofilm inoculum, nutrients and metabolism [5, 6]; metabolic responses to antibiotic treatments [7, 8] and to track cellulolytic activity [9]. This design has also been adapted to a closed-loop $CO_2$ accumulation system to monitor low-level microbial metabolic patterns during desiccation [10].

Understanding microbial activity at community and species levels does require the genetic, proteomic and metabolic profiling of the specific active and dormant organisms within an ecosystem. However, the traditional narrow focus on pathogenic species in water quality monitoring may well contribute to the many failures in detecting outbreaks, as indigenous communities could mask pathogens in our efforts to detect them. There are practical benefits and unique perspectives facilitated by the immediate and high-level availability of whole-ecosystem metabolic footprints. This is particularly true when coupled with modelling, which can predict many of the physico-chemical $CO_2$ fluctuations, resolving the biological data from total $CO_2$ information.

We evaluate the potential of using CEMS as an alarm in commercial and industrial settings, analogous to the miner's canary. The technology is assessed as an indicator in two similar but converse undesirable industrial scenarios: (1) the contamination of potable water systems, and (2) the inhibition of bioreactors for the treatment of waste- or industrial service water. The second scenario is especially relevant, since failure of wastewater treatment systems can result in serious environmental pollution, due to unexpected inflow of chemicals that inhibit growth, or careless incorporation of industrial waste streams without taking biodegradability into account. For instance, wastewater loading impact is commonly calculated based on COD rather than contaminant toxicity.

Any fluctuations of $CO_2$ production above a pre-determined minimum respiration rate can act as a high-level indicator of microbial contamination events in potable water systems, providing maintenance staff a warning to investigate more fully. Whether water is stored in a tank or distributed in pipes in systems such as dental chairs, microbial contamination of potable water is often the cause of disease outbreaks [11, 12]. With growing water scarcity challenges in drought-ridden areas, and the associated popularity of decentralized water systems gaining traction particularly in developing countries, remote monitoring for microbial contamination is paramount. Moreover, the microbiological monitoring of drinking water is limited by time (culturing), cost and expertise (quantitative real-time PCR, enzyme assays), which hampers the public provision of decentralized point-of-use water treatment systems. Most of these techniques involve the mass use of disposable reagents and equipment, a critical consideration in the current environmental waste crisis. CEMS has the benefit of real-time and remote monitoring of microbial activity, with equipment that is long-lasting and energetically sustainable using solar power, generally liberally available in drought-ridden areas.

In the converse scenario, in the biological treatment of sewage and industrial wastewater, microbial activity is responsible for converting dissolved pollutants into the gas phase, solid phase for removal as biomass, or into more innocuous chemical forms. Their metabolic activity can simultaneously act as a reliable indicator of changes in the waste treatment process, including changes in chemical composition (nutrient spikes, toxins, pH), temperature and failure of mixing (oxygenation vs anaerobiosis). Optimal functionality should be associated with a quantifiable $CO_2$ steady-state, and predictable fluctuation thresholds. Companies that utilize waste treatment bioreactors can be fined extensively if the reactors fail and toxic waste is released into water sources, recorded in governmental non-compliance records [13–15].

These fines are proportional to waste volume, and an immediate, durable indicator of bioreactor failure due to the disruption of microbial metabolism is both environmentally and financially valuable.

In this study, the thresholds of this $CO_2$ monitoring technology were shown to limit use as an early-warning system for potable water, but were sufficient to meet agricultural water standards. The system was an effective alarm for bioreactor disturbances. Modelling was employed to tease out physico-chemical $CO_2$ fluctuations from the metabolic response of the microbial aggregate, which is the active role player in waste remediation.

## Materials and methods

### Sampling: River water and wastewater

The CEMS performance was evaluated in two different contexts: for the monitoring of clean water, which should ideally contain very low levels of microbial contamination, and the monitoring of an active bioreactor. To assess the former, river water samples were collected from the polluted Plankenbrug River, Stellenbosch, South Africa (-33.933983; 18.85095) [16]. This polluted river water was diluted with pure water to mimic a range of pollution levels, from low to high. For the latter, Return Activated Sludge (RAS) samples were collected from a municipal wastewater treatment works in Cape Town, South Africa.

### Canary CEMS for potable water storage systems

Sterile 5 L Erlenmeyer flasks were filled with river water, mixed continuously to ensure aeration, and no additional nutrients added. The flasks were sealed and the CEMS tube for headspace gaseous analysis was placed directly above the liquid surface. A small air inlet port prevented negative pressure. The headspace gas was transferred via a peristaltic pump (flow rate = 15 mL/min; carrier gas = ambient air) through a LiCor $CO_2$ analyzer (Campbell Scientific, Stellenbosch, South Africa), with continuous data logging per second (Fig 1 –open loop). Thus, $CO_2$ measurements were investigated as a difference between atmospheric and reactor $CO_2$ levels, rather than absolute $CO_2$ production, as measured in $CO_2$-free air in previously described CEMS studies [5]. Atmospheric baselines were measured repeatedly on different days (10 days, minimum of 5 hrs per baseline), to assess variation. Two sensors were used throughout the experiment, calibrated every 3–7 days with a $CO_2$ gas standard (AFROX, Cape Town, South Africa), and randomly exchanged between control and experimental reactors throughout the study. The random sensor exchange was to monitor potential sensor drift and for normalizing, preventing the constant exposure of each sensor to either high or low $CO_2$

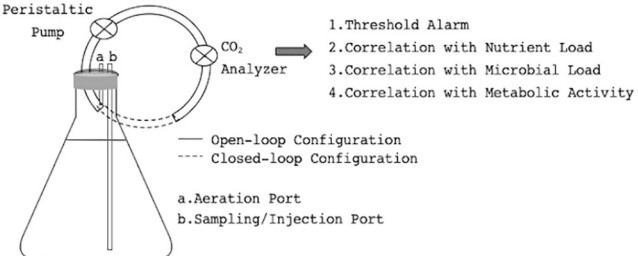

**Fig 1. Schematic of the laboratory-scale canary CEMS alarm system.** To assess minimum thresholds for whole-reactor CO2 monitoring, as an indicator of (1) water storage contamination events and (2) metabolic disturbances of bioreactors.

concentrations. Averages and standard deviations were calculated in Microsoft Excel for Mac (v 15.25), throughout the study.

**Headspace effect.** The effect of headspace volume on CEMS sensitivity was evaluated at different volumes: (1) 0.9 L headspace (5.0 L liquid), (2) 0.55 L headspace (5.35 L liquid) and (3) 0.25 L headspace (5.65 L liquid), as described in Table A1 in the S1 Appendix. In order to assess lower detection thresholds, both undiluted and 25% samples (v river water/v sterile tap water) were evaluated. The average total microbial $CO_2$ production [ppm $CO_2$ –ppm $CO_2$ (ambient control)] was plotted against headspace volume. Liquid surface area was recorded per treatment. The control was an identical reactor containing sterile (autoclaved, 121 ˚C, 15 psi, 30 min) river water. All further measurements were assessed at 0.9 L headspace (5.0 L liquid). The differences between means were compared individually at decreasing headspaces, between (1) & (2), and (2) & (3), for diluted and undiluted reactors.

**Minimum detection threshold.** In order to correlate $CO_2$ accumulation and SANS241 parameters [17], headspace accumulation of $CO_2$ in the river water reactors was measured (0.9 L headspace), along with the chemical and microbiological profiles (described below). To determine minimum thresholds, river water samples were subsequently diluted with sterile tap water (0%, 1%, 10%, 25%, 50%, 100% (v river water/v sterile tap water)), representing an increasingly contaminated potable water system, and evaluated for the same metabolic, chemical and microbiological profiles. Measurements were done for separate, triplicate experiments. These were assessed for linear correlations in Microsoft Excel for Mac (v 15.25) goodness-of-fit reported as $R^2$, assessed for significance at a 95% fit. These were plotted alongside the South African water quality guideline's chemical and microbiological limits for application in monitoring potable water storage [17]. National guidelines are based on the World Health Organization guidelines [18], and comparable to those of leading countries like Canada [19].

**Chemical profile of river water samples.** Chemical Oxygen Demand (Spectroquant COD Cell Test, 1.14541.0001), Nitrogen (Spectroquant Nitrogen (total N) Cell Test, 1.00613.0001), Ammonium (Spectroquant Ammonium $NH_4^+$ Test, 1.14752.0001), Sulphate (Spectroquant Sulfate $SO_4^{2-}$ Cell Test, 1.14548.0001) and Phosphate (Spectroquant Phosphate Test, 1.14842.0001) were measured colorimetrically using a Merck Spectroquant Pharo300 photometer, according to the manufacturer's instructions. Samples were heated in a Spectroquant TR320 Thermocycler, thoroughly mixed by inversion before analyzing and filtered with cellulose nitrate, 1.2 μm pore size filters (Sartorius Stedim Biotech, South Africa). Spectroquant kits and apparatus were sourced from Merck (Modderfontein, South Africa). Correlations between chemical profiles and headspace detection of $CO_2$ were determined in triplicate against linear regression models in Microsoft Excel for Mac (v 15.25), and goodness-of-fit reported as $R^2$, assessed for significance at a 95% fit.

**Microbiological profile of river water samples.** Standard Heterotrophic Plate Counts were conducted (APHA Method 9215) on Standard Plate Count media (Yeast Extract, 2.5 g/L; Pancreatic Digest of Casein, 5.0 g; Glucose, 1.0 g/L; Agar, 15 g/L; pH 7.0; 26˚C). Total coliform counts were assessed on EndoAgar (Merck, APHA Filtration Method 9222; 37˚C). Gram negative enteric bacteria were assessed on MacConkey Agar (APHA Filtration Method 9222; 37˚C). In addition to filtration, where bacterial loads were higher, reactor samples were diluted in Physiological Saline Solution (9 g/L NaCl), and enumerated on Standard Plate Count, Endo and MacConkey Agar. Chemicals were purchased from Sigma-Aldrich (Johannesburg, South Africa). Correlations between microbiological profiles and headspace detection of $CO_2$ were determined in triplicate against linear regression models in Microsoft Excel for Mac (v 15.25), and goodness-of-fit reported as $R^2$, assessed for significance at a 95% fit.

**ATP measurements.** ATP [Relative Light Units (RLU)/50 μL] was measured as a confirmation of metabolic profiles. River water was added (50 μL) to Hygenia Ultrasnap ATP surface

swabs (Fischer Scientific, Ottawa, Canada), shaken for 15 s and measured in a Hygenia EnSURE luminometer according to manufacturer's instructions. These are used to measure microbial surface contamination in the food and hospital industry [20], but were harnessed here as a simple tool for confirming metabolic activity in a river water dilution series. Dispersion of flocs was critical for unbiased readings. Samples (1.5 mL) were vortexed (2 minutes) prior to analysis. Correlations between metabolic profiles and headspace detection of $CO_2$ were determined in triplicate against linear regression models in Microsoft Excel for Mac (v 15.25), and goodness-of-fit reported as $R^2$, assessed for significance at a 95% fit.

**Closed-loop design.** To investigate lower detection thresholds, the system was redesigned to accumulate $CO_2$ over time (Fig 1 –closed loop), by circulating the reactor headspace continuously over the $CO_2$ analyzer. An overnight culture of *E. coli* (3 g/L Tryptic Soy Broth, TSB, 26 ˚C, rotary shaker) was diluted (sterile tap water) and inoculated into a 5 L Erlenmeyer flask containing autoclaved tap water, at final concentrations of $10^1$, $10^2$ and $10^3$ CFU/100 mL, with continuous stirring. Overnight *E. coli* concentrations were pre-determined with growth curves at room temperature rather than 37 ˚C in TSB, to prevent temperature variation upon transfer. Final cell concentrations were checked by extraction from the sampling port, and counted on Tryptic Soy Agar using filtration (TSA, 3 g/L TSB, 15 g/L agar), as described above for total heterotrophic analysis. The accumulation of $CO_2$ over time was measured at each cell concentration, before the cell concentration was increased via the injection port.

## Canary CEMS for active bioreactor disturbances

For RAS bioreactors, the flasks were filled (4.5 L) with synthetic wastewater and sterilized by autoclaving. The reactors were inoculated with 0.5 L activated sludge, corresponding to a liquid volume of 5L and a headspace volume of 0.9 L, and allowed to equilibrate to steady state $CO_2$ production with constant stirring. One of the two reactors was sterilized in the autoclave after the addition of RAS and maintained under identical conditions as an abiotic control. The reactors were treated with identical environmental stressors. These chemical and physical disturbances included (1) chlorine [ChlorGuard commercial bleach; final reactor concentration 15% (v/v)], (2) acidification [(a) pH 6.6 dropped to pH 5.0, ~200 μL concentrated (38%) HCl and (b) pH 6.6 dropped to pH 3.3, ~600 μL concentrated HCl)] and (3) temperature fluctuations [23˚C dropped to 16˚C; 5% (v/v) ice, in a 4 ˚C refrigerator]. The chlorine treatment was repeated, once with reactor mixing (aerated) and once without (sedimentary). For the pH disturbances, soluble $CO_2$ was measured pre- and post-acidification with a MerckMillipore Carbon Dioxide Titrimetric MCarbon Test (Modderfontein, South Africa), according to manufacturer's instructions. A Jenway 3510 pH Meter, calibrated using the 3 buffer method, was used to monitor pH. Titrations were performed and data collected using a HI902 Auto-titrator (Hanna Instruments®) equipped with a HI1131 pH probe (Hanna Instruments®). The instrument was pH calibrated beforehand using standards pH 4, pH 7 and pH 10 (Merck Millipore). Up titrations were performed to endpoint pH 11 using standardized 0.1 N NaOH after which subsequent down titrations were performed with standardized 0.1 N HCl to endpoint pH 5.

For all treatments, cell concentrations were measured pre- and post-disturbance. Standard plate counts were assessed with dilutions on Tryptic Soy Agar (Sigma Aldrich, Johannesburg, South Africa).

**Mathematical modelling of physico-chemical and microbiological $CO_2$ release.** A dynamic mathematical model (the details of which can be found in S1 Appendix) was used to discriminate between physico-chemical- and microbiological $CO_2$ release from the liquid. A fundamental assumption of this model was that the microbiological $CO_2$ production was

constant. While this assumption may appear limiting, it has little effect on the results interpretation, as demonstrated shortly.

The model accounted for microbiologically produced $CO_2$, speciation into bicarbonate ($HCO_3^-$ and carbonate $CO_3^{2-}$), and $CO_2$ mass transfer between the liquid- and gas phases. The model was implemented using MATLAB R2018a (The MathWorks, Inc., Natick, MA).

All model parameters are provided in Table A1 in S1 Appendix. Two parameters were unknown: the microbiological $CO_2$ production rate ($\dot{r}_{CO2}$) and the liquid-air mass transfer coefficient ($K_LA$). Both were determined by regression of experimentally measured gaseous $CO_2$ concentrations to model predictions.

**Threshold alarm software.** Remote alarm functionality was achieved by developing an external software application in the Python programming language to intercept the data packets on the LiCor LI-820 $CO_2$ analyzer's serial port. This allowed users to specify an upper and lower threshold, and to configure an email message, sent when serial port readings surpassed any of the pre-determined thresholds.

## Results

The measurement of microbial $CO_2$ production for the remote monitoring of water treatment system performance was assessed. Two experimental scenarios were evaluated, on the polar ends of a metabolic spectrum: (a) for the remote monitoring of microbial contamination in potable water systems (where the ideal microbial levels are zero), and (b) for the remote monitoring of steady state microbial activity in bioreactors for waste treatment (where steady-state $CO_2$ production is optimal within a pre-determined window).

### Headspace volume and detection sensitivity

Respiration was monitored in reactor headspaces, which were open to the atmosphere via a defined inlet port. Baselines were predictable at 385 ppm $CO_2$ ± 45 ppm for the open-loop system, with variation attributed to human activity in the vicinity increasing $CO_2$ levels during the day. Closed-loop baselines had a similar standard deviation if measured at separate instances over different days, but a variation of less than 5 ppm over days if sealed. Headspace $CO_2$ was allowed to accumulate before open-loop measurement. In river water reactor systems, a smaller headspace decreased the sensitivity of the whole-system $CO_2$ production (Fig 2, Student's t-Test, p<0.05). This was counter-intuitive, as lower headspace volumes are correlated with greater river water volumes, increasing the total biomass and thus presumably resulting in increased $CO_2$ production. These unexpected results are likely due to two reasons: (1) the system is open to ambient air and a smaller headspace is more rapidly replaced by ambient air drawn into the system, and (2) the conical flasks in the laboratory set-up meant a larger headspace was equivalent to a larger surface area for gas-liquid $CO_2$ exchange. The tension between the influences of these design parameters on the data indicates that case-by-case optimization is necessary for effective industrial CEMS application as an early-warning alarm system for microbial disturbances in water storage systems.

The dynamic model results compare well to the experimentally measured $CO_2$ partial pressure (Fig 3a), but the rate of $CO_2$ production (as determined by regression) was so low in the river water that the regression was insensitive to orders of magnitude variations in $\dot{r}_{CO2}$ in the open-loop configuration, which is to be expected with only a few cells responsible for the metabolic footprint. Fig 3b shows the variation in the coefficient of determination (1-TSS/RSS, where TSS is the total sum of square errors from the mean, and RSS is the residual sum of squares) as a function of the regressed parameters $K_LA$ and $\dot{r}_{CO2}$. Clearly, the values for $\dot{r}_{CO2}$

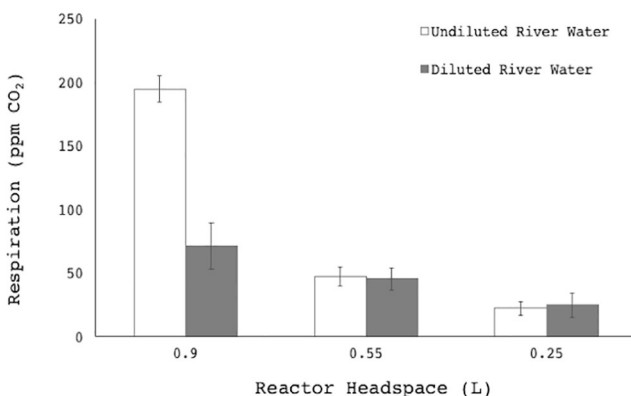

**Fig 2. The effect of reactor volume and headspace on (a) the whole-system $CO_2$ measurement.** The first reactor (1) contained undiluted river water at (A) 5 L with 0.9 L headspace, (B) 5.35 L with 0.55 L headspace, and (C) 5.65 L with 0.25 L headspace. The second reactor (2) contained diluted Plankenbrug river water (25% v/v sterile tap water), at the same volume:headspace ratios. Controls were a separate reactor with undiluted autoclaved Plankenbrug river water. Respiration was compared by subtracting the average $CO_2$ production of the sterile control reactor from the average $CO_2$ production of the reactor over a given time period. Error bars represent standard deviation of measurements per second over 1 hour.

can vary significantly without influencing the fraction of explained variance. This implies that the CEMS device is not suitable for such low $CO_2$ production rates in an open-loop design. However, the combination of modelling and experimentation as performed in this study can be used to indicate the range at which the system can be used, guiding further modification of the system into a closed-loop design as described below.

The open-loop CEMS data proved insufficient to accurately predict the rate of $CO_2$ production in systems with low levels of microbial contamination. However, the steady state $CO_2$ concentrations in such systems were still significantly higher than in their abiotic counterparts. This implies that the steady state $CO_2$ concentration as detected by the CEMS can be used as an empirical predictor of water quality. Given the enhanced sensitivity at larger headspace volumes, correlating to larger surface area, a liquid volume of 5 L (corresponding to a headspace volume of 0.90 L) was used for all subsequent experiments.

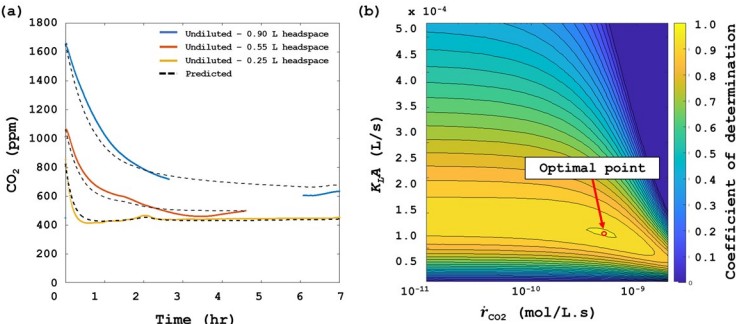

**Fig 3. Modelling of $CO_2$ limits of detection.** (a) Predicted vs measured values of $CO_2$ partial pressure in the headspace above undiluted river water for varying headspace volumes. $CO_2$ was allowed to accumulate in the headspace prior to turning on the CEMS. (b) Residual Sum of Squares as a function of the regressed parameters $\dot{r}_{CO2}$ and $K_LA$. At low contamination levels, the rate of $CO_2$ production can vary by multiple orders of magnitude without significantly affecting the measured response, thereby indicating the CEMS limits of detection.

## Monitoring microbial contamination: Potable water systems

Elevated $CO_2$ concentrations in the reactor headspace was measurable for undiluted polluted river water in an open system. In order to determine the sensitivity thresholds of the system, the river water was diluted (50%, 25%, 10%, 1%, 0% v/v) with sterile tap water, representing an increasingly contaminated water storage system, and the profiles were assessed at each dilution.

- chemical (COD, Total-N, $NH_4^+$-N, $SO_4^-$),

- microbiological (standard heterotrophic bacteria, total coliforms, enteric bacteria), and

- metabolic ($CO_2$, ATP)

The dilution profile of polluted river water produced a measurable $CO_2$ production trend, which was correlated to a suite of water quality parameters, described in the national water quality guidelines [17]. Adenosine triphosphate (ATP) was included as an independent measure of metabolic activity, detected with fluorometry on swabs typically utilized in surface metabolic monitoring. This technique confirmed the metabolic profile with tight correlation to $CO_2$ data ($R^2 = 0.832$; Fig 4b), but generated high variation at higher bacterial loads, likely due to challenges in microbial floc dispersion. All of the chemical and microbiological parameters also correlated with $CO_2$ production (Fig 4b and 4c), and the point at which the detection system's correlation with the chemical parameters (total N, $NH_4$-N and sulphate) was useful fell above the minimum thresholds according to South African Water Quality Guidelines [17]. In contrast, the limits prescribed by these guidelines for COD and the microbial loads were well below the lowest threshold of the system. The correlation ($R^2 = 0.88$; Fig 4) between enteric bacterial load and $CO_2$ production indicated promising potential for detection of larger contamination events. However, the target water quality range for total heterotrophs, according to SANS241 [17], is 0–100 CFU/mL, which is below the lowest threshold of the $CO_2$ correlation in open-loop configuration, and the target water quality range for coliforms is drastically more stringent at 0–5 CFU/100 mL.

Based on these threshold limitations of the open-loop CEMS alarm system, measuring $CO_2$ production on a continuously replaced carrier gas, the design was adapted to measure closed-loop $CO_2$ accumulation, in an attempt to decrease the lower threshold (Fig 1). It was only assessed against the microbial load, which is the limiting parameter (Fig 4). In this case, the rate of $CO_2$ accumulation, calculated from the slopes, was compared (ppm/h) at 10 CFU/100 mL, $10^2$ CFU/100 mL and $10^3$ CFU/100 mL (total coliforms). Here, the $CO_2$ accumulation rates were statistically distinguishable as low as 10 CFU/100mL according to linear best fit models, however the tool is truly useful over 100 cfu/100 mL, as the slopes at 10 and 100 cfu/ mL only differ by 5 ppm, and control variation is 1.8 ppm (Fig 5). An automated flushing mechanism to facilitate closed-loop daily monitoring has been designed in-house, to allow for a pre-determined window of $CO_2$ accumulation before flushing, circumventing perpetual $CO_2$ accumulation.

It is clear from these thresholds that the open-loop Canary CEMS is an indicator of significant, sudden contamination events, but is not suited for the monitoring of drinking water to potable national and international standards. The lack of suitability is based largely on sensitivity of the system for measuring microbial activity at levels as low as guidelines demand, which is effectively zero. The alarm effectively detects microbial loads higher than $10^2$–$10^3$ CFU/mL, above the maximum thresholds for potable water. In attempting to compare the detection range to other widely promoted measurements techniques, such as qPCR, the definition of Limits of Quantification (LoQ's) are informative, including precision and accuracy, only

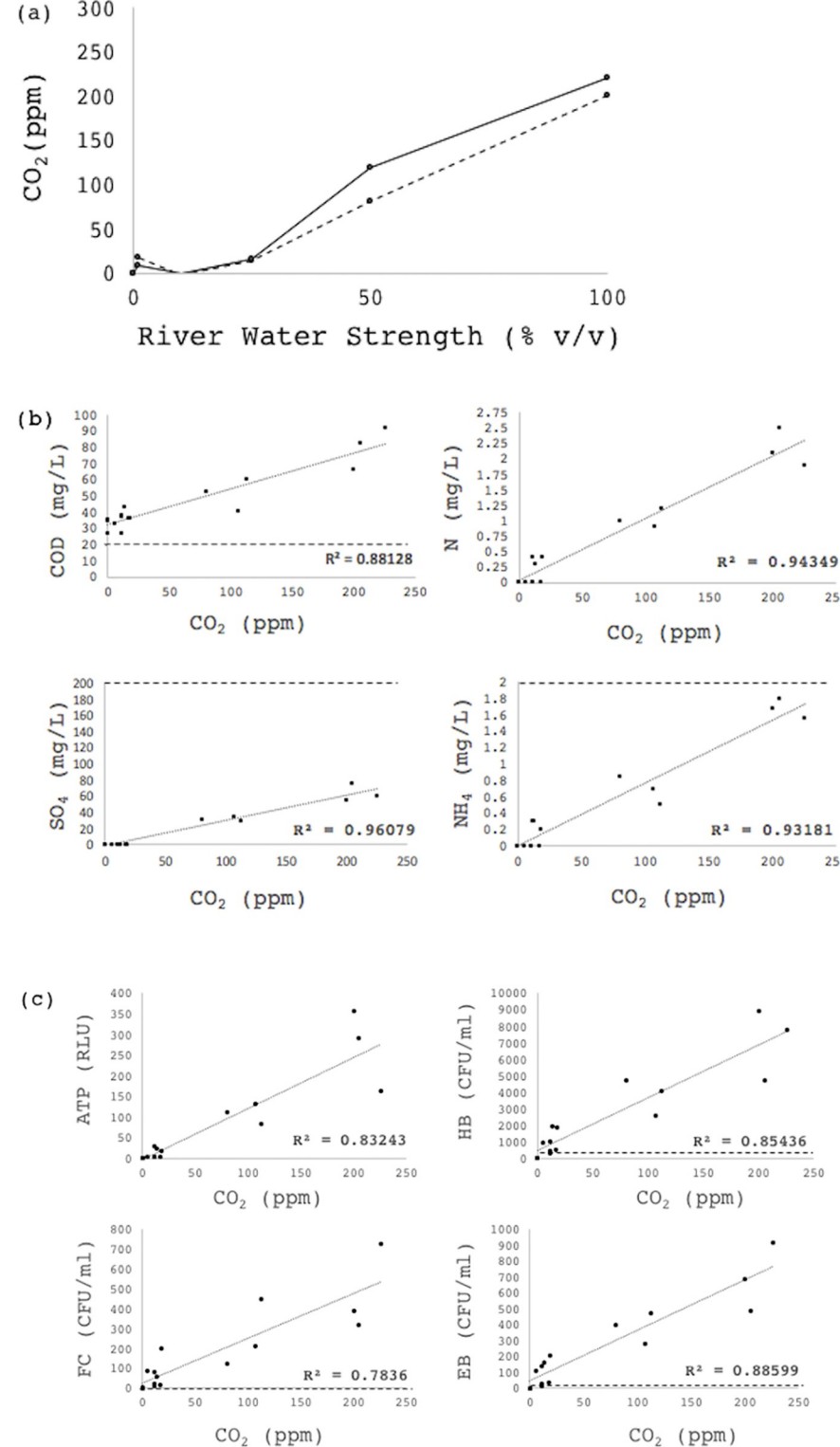

**Fig 4.** (a) Pollutant effects on $CO_2$ measurements. $CO_2$ profiles for increasing river water concentrations (pollution levels) were plotted for 2 independent sampling events. (b) Pollutant effects on $CO_2$ measurements. River water chemistry was correlated against $CO_2$ at the above-mentioned dilutions, including COD (mg/L), sulphate (mg-SO4-/L), total nitrogen (mg-N/L) and ammonia (mg-NH4+/L). The results of triplicate experiments are plotted. Minimum thresholds are indicated with dashed lines, as determined by the South African Guidelines for Drinking Water

Standards [17]. (c) River water energetic (ATP) and microbiological profiles were correlated against $CO_2$ at the above-mentioned dilutions, including fecal coliforms (FC), enteric bacteria (BC) and heterotrophic bacteria (HB). The results of triplicate experiments are plotted. Minimum thresholds are indicated in red, as determined by the South African Guidelines for Drinking Water Standards [17].

under *stated experimental conditions* [21]. Forootan et al. [22] explore the challenges of defining thresholds of detection and quantification of qPCR. In their context, the LoD (Limit of Detection) was 2 molecules, and the LoQ was 16 molecules. This is tenfold lower than the CEMS alarm, but still demands time, laboratory facilities and expertise.

Contamination events that have warranted reporting due to associated public illness include ranges as wide as 20 CFU/mL [23] to 10300 CFU/mL [24]. This work demonstrates the utility of the technology as an early warning system for such significant contamination events, as well as for applications in aquaculture, irrigation (water quality limited to 100 CFU/mL, according to South African National Guidelines [25]) and service water for washing equipment.

## Monitoring bioreactor disturbances

The alarm system was proven effective as an indicator of microbial inhibition and metabolic disturbances in active return activated sludge (RAS) reactors. In the case of a disinfectant (chlorine, Fig 6a and 6b), a pH disturbance (pH drop from 6.6 to 5.0 and 3.3, HCl, Fig 6c & 6d respectively) and a temperature disturbance (temperature decrease: 22˚C to 16˚C, Fig 6e), the $CO_2$ immediately fluctuated well outside of pre-determined windows defined by steady-state $CO_2$ measurements.

The first disinfectant treatment triggered an approximately 5-fold increase in the metabolic activity of a mixed reactor (Fig 6a). The $CO_2$ measurements subsequently steadily decreased over days, remaining almost 2-fold higher than the pre-treated whole-reactor $CO_2$ production rate. The control reactor showed a slight increase in $CO_2$ production upon addition of chlorine, probably due to chemical oxidation of organic material in the water, although the metabolic measurements were negligible in comparison to the non-sterile reactor confirming the biological metabolic response (Fig 6b). The sedimentary reactor demonstrated a similar, although less dramatic, response upon disinfectant treatment, suggesting a higher sedimentary

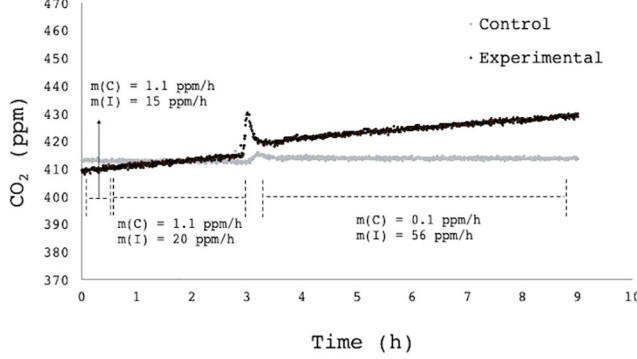

**Fig 5. Headspace accumulation of CO2 in in a closed-loop CEMS design.** A control (C) reactor containing sterile water was compared to *E. coli* inoculated (I) reactors. The inoculation concentration was increased twice: from an initial concentration of 10 CFU/mL over time range (A), it was increased to 100 CFU/mL for time (B) and finally (C) 1000 CFU/mL. The $CO_2$ generation rates m(C) and m(I) in the control and inoculated reactors, respectively, are indicated in the figure, clearly showing a detectable difference in each case.

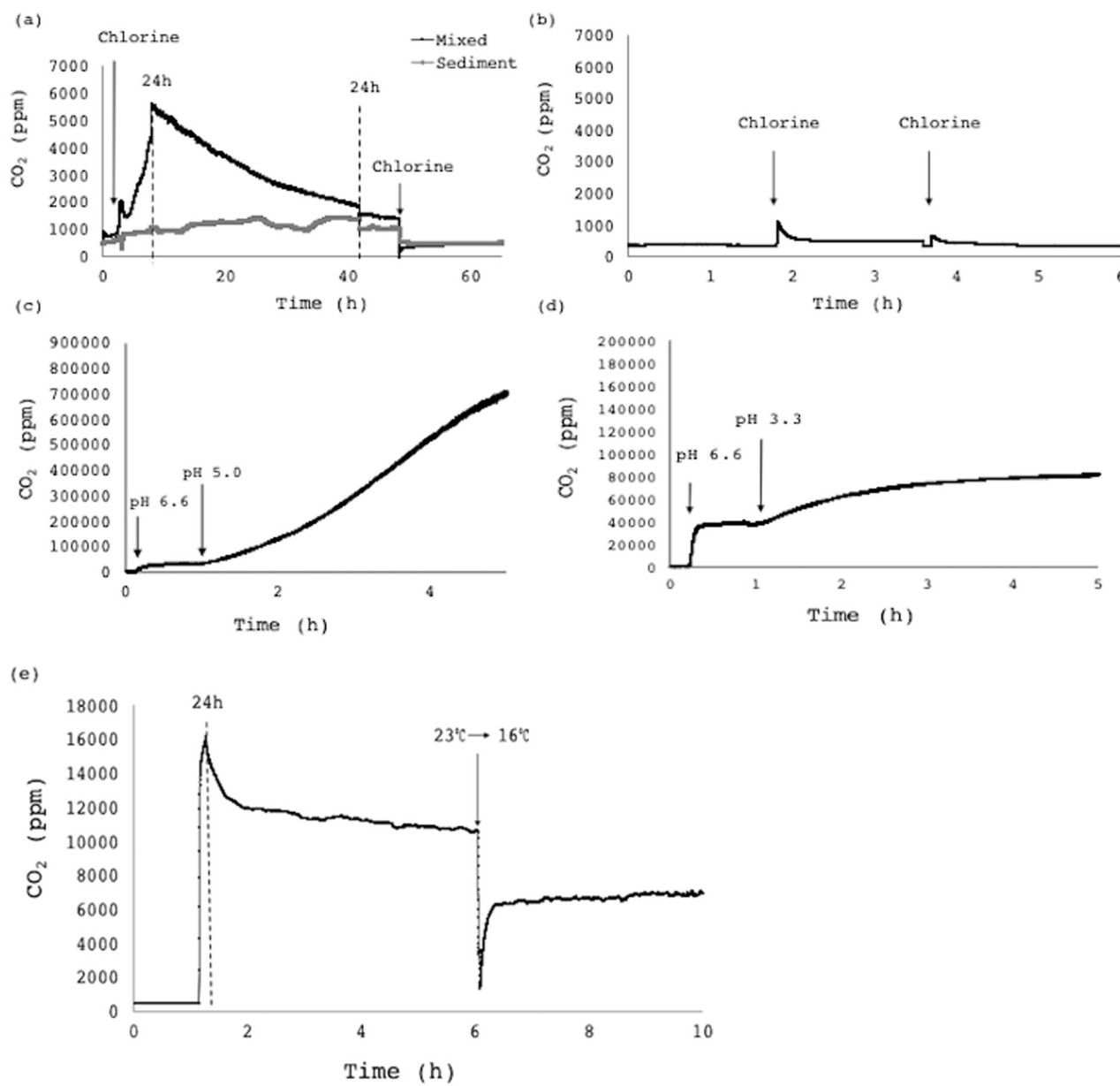

**Fig 6. $CO_2$ fluctuations in response to physico-chemical disturbances.** The $CO_2$ profiles of (a) a mixed and sedimentary RAS reactor with the addition of chlorine compared to (b) an autoclave sterilized RAS control reactor; as well as a mixed RAS reactor acidified from pH 6.6 to (c) 5.0 and (d) 3.3, with concentrated HCl, and (e) a sudden cooling event (ice; ~5% v/v) in a mixed RAS reactor connected to the system at 1 h. Measurement intervals are indicated with dashed lines.

chlorine demand. A second treatment with the same disinfectant concentration dropped the measurable $CO_2$ production rate to zero for both the mixed and sedimentary reactors, and cell survival was also approximately zero after the second chlorine treatment. It is well established that microbes respond to stressors such as antibiotic challenges with a spike in $CO_2$ production, likely due to the metabolic shifts for adaptation [7, 8], particularly with continued exposure to the chemical inhibitor. Higher $CO_2$ production is potentially ascribed to the metabolic cost associated with the upregulation of efflux pump activity in the cell [7]. This work indicates a similar metabolic response to an oxidizing agent (Fig 6a).

**Table 1. Ratio of $CO_2$ partial pressures before and after acidification due to purely chemical effects (calculated) compared to combined chemical and metabolic effects (measured).**

| Initial pH | Final pH | $\frac{p_{CO2,f}}{p_{CO2,i}}$ (calculated) | $\frac{p_{CO2,f}}{p_{CO2,i}}$ (measured) |
|:---:|:---:|:---:|:---:|
| 6.6 | 5.0 | 2.5 | 23.9 |
| 6.6 | 3.3 | 2.7 | 2.3 |

Similarly, a pH decrease from 6.6 caused the microbial consortium to increase its metabolic rate (Fig 6c and 6d). The measurable metabolic rate was approximately 10-fold higher at pH 5.0 than at pH 3.3. Acidification to pH 3.3 caused a drop in cell concentration from $10^9$ CFU/mL to $10^7$ CFU/mL in 2 h and to $10^3$ CFU/mL in 24 h, whereas cell concentrations remained constant at $10^9$ CFU/mL with a milder pH drop to 5.5. The 10-fold difference in $CO_2$ production is due to an interaction between this difference in biological contribution, as well as physico-chemical $CO_2$ speciation.

Temperature and pH affect the liquid phase speciation of $CO_2$ (see S1 Appendix) and will affect the liquid-to-headspace mass transfer rate. A decrease in pH shifts the carbonate equilibria from ionic bicarbonate towards dissolved $CO_2$, which would then escape to the atmosphere and subsequently increase the headspace $CO_2$ concentration. Assuming the liquid- and gas phases are in equilibrium and that the acidification process is rapid enough such that the total (combined liquid phase and headspace) molar amount of $CO_2$ in the reactor remains constant, it is shown that the proportional increase in $CO_2$ partial pressure $p_{CO2}$ is given by Eq 1 (subscripts 0 and $f$ indicate reactor conditions before and after acidification, see S1 Appendix):

$$\frac{p_{CO2,f}}{p_{CO2,0}} = \frac{\frac{V_G}{RT_0} + \frac{V_L}{K_H(T_0)}\left(1 + \frac{K_1(T_0)}{10^{-pH_0}} + \frac{K_1(T_0)K_2(T_0)}{10^{-2pH_0}}\right)}{\frac{V_G}{RT_f} + \frac{V_L}{K_H(T_f)}\left(1 + \frac{K_1(T_f)}{10^{-pH_f}} + \frac{K_1(T_f)K_2(T_f)}{10^{-2pH_f}}\right)} \quad (1)$$

Where $V_G$ and $V_L$ are the gas- and liquid-phase volumes, $R$ is the universal gas constant, and $K_H$, $K_1$ and $K_2$ are the Henry's constant and dissociation constants of dissolved $CO_2$. The equilibrium constants are all functions of temperature.

Within this study, the ratio of steady state $CO_2$ partial pressures (before and after acidification) due to purely chemical effects, calculated using Eq 1, is compared to the ratio of steady state $CO_2$ partial pressures as measured using the CEMS (Table 1). When the pH was dropped to a value of 5.0 from pH 6.6, the measured 24-fold increase in $CO_2$ partial pressure far exceeded the values solely attributed to a disturbance in the chemical equilibrium, thus indicating that acidification triggered a metabolic response. In the case of acidification to pH 3.3, the measured 2.3-fold increase in $CO_2$ partial pressure was slightly lower compared to that calculated for a purely chemical response. This indicates a complete cessation of metabolic activity. The lower than expected increase in $CO_2$ partial pressure can be attributed to the continuous removal of $CO_2$ from the reactor headspace by the CEMS.

In contrast, temperature fluctuations did not lead to a spike in $CO_2$ production rates, but rather a drop in respiration to approximately 65% upon cooling (Fig 6e). There was no associated decrease in microbial load with the temperature fluctuation. The phenomenon of decreased respiration with decreasing temperature is well-demonstrated in soil microbial communities [26].

The temperature decrease, from 22 ˚C to 16 ˚C, would affect the chemical equilibrium mathematically represented by $K_H$, $K_1$ and $K_2$. In this case, the $CO_2$ partial pressure due to chemical effects alone was calculated to be 10%, underestimating the measured decrease of 40%, clearly indicating a decrease in metabolic activity.

The LiCor $CO_2$ monitoring system software was also adapted to send an email in response to $CO_2$ fluctuations outside pre-determined thresholds, alerting the system manager of metabolic disturbances, remotely and immediately. This adaptation eliminates the need for expertise to assess microbial activity, and the consequent performance of the bioreactor. It makes it accessible to technicians, which is critical in implementing this technology, particularly in rural, under-developed areas. Design adjustments might involve the circulation of a representative reactor sample through an adjacent compartment controlled for volume and headspace, rather than direct measurement of the reactor headspace, as well as the measurement of biofilms in pipes.

There is no silver bullet for the monitoring of water storage contamination events, be it potable, irrigation or water recirculation for cleaning—a common practice in drought-ridden areas. The high frequency of water quality system collapse demands creative additions to the current suite of water-monitoring techniques. There is a tension between the resolution of microbial water quality information, the cost and the speed at which information is accessible.

Thus, the efficiency of this technology can be compared to competing technology [27]. Such competing technologies, based on fluorescence assays and enzyme detection, still involve grab samples, and demand technical know-how and complex, energy-expensive technology. However, the true benefit of this work lies not in competition, but rather in the amplified benefits of multipronged approaches to water quality monitoring. The cumulative effect of both high-level, remote, online alarm systems and more detailed pathogen identification, will provide more effective protection to communities susceptible to outbreaks. In addition, the particular benefit of this technology is the accessibility to under-developed areas without geographical and economic access to standard laboratory facilities: often the most vulnerable communities.

## Conclusions

The Carbon Dioxide Evolution Monitoring System was shown to only be indicative of relatively high levels of microbial contamination in water systems. The lower thresholds are above the maximum coliform and heterotrophic loads permitted for potable drinking water according to WHO and national standards, although significantly improved by the redesign to a closed-loop system. In contrast to potable water standards, it was proven effective for significant contamination events and for monitoring water for broader applications, for example to national and international irrigation standards. The canary CEMS was also effective as an immediate, remote alarm for metabolic disturbances in a RAS reactor treated with a disinfectant, as well as pH and temperature fluctuations. The aim of the system is to act as an immediate indicator of the metabolic responses of the microbial aggregate, which was resolved from the physico-chemical $CO_2$ equilibration by mathematical modelling. If thresholds and minimum limits are understood and optimized for individual application per system, the universal principle of whole microbiome respiration has notable potential as an early-warning technology for microbial disturbances in industrial settings.

## Supporting information

**S1 Appendix. Mathematical model of $CO_2$ release.**
(DOCX)

## Acknowledgments

The authors are grateful to Jaco van Rooyen (Department of Process Engineering, Stellenbosch University) for his assistance with pH auto-titrations.

## Author Contributions

**Conceptualization:** Gideon M. Wolfaardt.

**Data curation:** Wendy Stone.

**Formal analysis:** Wendy Stone, Tobi M. Louw.

**Funding acquisition:** Gideon M. Wolfaardt.

**Investigation:** Wendy Stone.

**Methodology:** Wendy Stone, Tobi M. Louw, Marthinus J. Booysen.

**Project administration:** Wendy Stone.

**Resources:** Wendy Stone, Marthinus J. Booysen.

**Software:** Marthinus J. Booysen.

**Supervision:** Gideon M. Wolfaardt.

**Validation:** Wendy Stone, Tobi M. Louw.

**Visualization:** Wendy Stone.

**Writing – original draft:** Wendy Stone.

**Writing – review & editing:** Tobi M. Louw, Marthinus J. Booysen, Gideon M. Wolfaardt.

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
