## [Decision Letter · Decision Letter 0]

19 Oct 2020

PONE-D-20-04971

Canary in the coliform mine: Exploring the industrial application limits of a microbial respiration alarm system

PLOS ONE

Dear Dr. Stone,

Thank you for submitting your manuscript to PLOS ONE. After careful consideration, we feel that it has merit but does not fully meet PLOS ONE’s publication criteria as it currently stands. Therefore, we invite you to submit a revised version of the manuscript that addresses the points raised during the review process.

We look forward to receiving your revised manuscript.

Kind regards,

Dawei Zhang, Ph.D.

Academic Editor

PLOS ONE

Journal Requirements:

Additional Editor Comments (if provided):

This manuscript describes an early-warning system for microbial disturbances in industrial settings by exploring microbial biofilm growth and metabolic responses. It is an interesting paper. However, the experiment design and statistical analysis need to be improved. I would like to receive the revised paper if you can address the reviewer's comments.

Reviewers' comments:

Reviewer's Responses to Questions

**Comments to the Author**

1. Is the manuscript technically sound, and do the data support the conclusions?

Reviewer #1: Partly

2. Has the statistical analysis been performed appropriately and rigorously? 

Reviewer #1: No

3. Have the authors made all data underlying the findings in their manuscript fully available?

Reviewer #1: Yes

4. Is the manuscript presented in an intelligible fashion and written in standard English?

Reviewer #1: Yes

5. Review Comments to the Author

Reviewer #1: This is an interesting paper and for the most part is carefully performed and well written. However, some of the statistical analyses and some figures don’t look correct to me and I am requesting a revision primarily to address this.

L122-125 It is stated that river water was used to assess “very low levels of microbial contamination”. However, the river is from a highly polluted source as stated earlier in the manuscript and as is supported by the chemical data that show very high COD. It needs to be explained here that the river water will be added in very low concentrations to pure water in order to mimic clean water.

L131-141 , 149-161

Its not stated explicitly how many sensors were used so I assume it was just one, and the sensor was used in sequential experiments to measure differences. Standard deviations were calculated from the STD of measurements obtained as a time series during a single sensor run.

The problem with this is that any temporal drift in the sensor output will be interpreted as a difference between treatments. It also violates the assumptions of a t-test which requires independent measurements when in this case, all data for the same treatment were autocorrelated. It is presumably this that accounts for the supposed significant differences between some treatments in Fig 2 despite near identical means. Instead, the figure should show the average and STD derived from repeated (independent) experiments with the treatments run in randomized order to counteract systematic biases caused by drift. If the experiment wasn’t repeated then variance cant be assessed.

L380 “ Headspace CO2 concentration had a linear correlation with the 380 dilution of the river water,” between 25 and 100% strength (Fig 4; R2 = 0.986) and all other data with R2 in Fig 4.

AND

L287 “Thistechnique confirmed the metabolic profile with tight correlation to CO2 data (R2=0.99; Fig 4),

The R2 values reported are much higher than would be expected from the scatter seen in the data in the figures. The only explanation I can think of is that the data were averaged before calculating R2, which is the difference between the two plots shown here: https://i.imgur.com/GDZmrjv.png . This artificially deflates variance by performing stats on already processed data.

L414 Fig 5 caption doesn’t match the figure. I don t understand what Fig 5 is showing.

6. PLOS authors have the option to publish the peer review history of their article (what does this mean?). If published, this will include your full peer review and any attached files.

Reviewer #1: No

---

## [Author Response · Author response to Decision Letter 0]

29 Nov 2020

Editor Journal Requirements:

and

Authors:

Thank you for the guidance. We have gone carefully through formatting, and have met the stipulated requirements. If further editing is necessary, we are happy to do so. 

As primary and corresponding author, I made the mistake of not using track changes as I edited. 

However, the only edits we made were those recommended by the reviewers and described below (including Line numbers of edits). In addition, I formatted according to the links provided (headings, figure and table captions, and references). All figures were corrected using PACE.

I highlighted the changes in yellow, to account for the mistake of not using track changes before saving the new, edited document.

Additional Editor Comments (if provided):

This manuscript describes an early-warning system for microbial disturbances in industrial settings by exploring microbial biofilm growth and metabolic responses. It is an interesting paper. However, the experiment design and statistical analysis need to be improved. I would like to receive the revised paper if you can address the reviewer's comments.

Authors:

We thank the editor for the opportunity to respond to the reviewer’s comments and resubmit. We hope we have addressed all comments to your satisfaction, and are available for follow up if further issues need clarification. 

The reviewer’s comments were astute, and we are grateful for the improvement of the manuscript through the editorial process.

Reviewer:

L122-125 It is stated that river water was used to assess “very low levels of microbial contamination”. However, the river is from a highly polluted source as stated earlier in the manuscript and as is supported by the chemical data that show very high COD. It needs to be explained here that the river water will be added in very low concentrations to pure water in order to mimic clean water.

Authors:

We thank the reviewer for this clarification, and have adjusted the sentence as follows: 

“This polluted river water was diluted with pure water to mimic a range of pollution levels, from low to high.” (Line 118-119)

Reviewer:

L131-141 , 149-161 Its not stated explicitly how many sensors were used so I assume it was just one, and the sensor was used in sequential experiments to measure differences. Standard deviations were calculated from the STD of measurements obtained as a time series during a single sensor run. 

The problem with this is that any temporal drift in the sensor output will be interpreted as a difference between treatments. It also violates the assumptions of a t-test which requires independent measurements when in this case, all data for the same treatment were autocorrelated. It is presumably this that accounts for the supposed significant differences between some treatments in Fig 2 despite near identical means. Instead, the figure should show the average and STD derived from repeated (independent) experiments with the treatments run in randomized order to counteract systematic biases caused by drift. If the experiment wasn’t repeated then variance cant be assessed.

Authors:

Two sensors were used, rather than one, but the reviewer is correct in this statistical observation. The sensors were regularly interchanged throughout all the runs, between control and experimental reactors, and between high and low CO2 accumulation, in order to assess drift. We can confirm that sensor drift is negligible. Both this random, regular exchange between sensors during experimental work confirms this, and, as stated earlier in the manuscript “Respiration was monitored in reactor headspaces, which were open to the atmosphere via a defined inlet port. Baselines were predictable at 385 ppm CO2 ± 45 ppm for the open-loop system, with variation attributed to human activity in the vicinity increasing CO2 levels during the day. Closed-loop baselines had a similar standard deviation if measured at separate instances over different days, but a variation of less than 5 ppm over days if sealed.” The low variance under similar controlled conditions on different days allows us to safely conclude that any sensor drift is negligible. We thank the viewer for this meticulous caution, and have added a sentence about using two sensors with random interchange during experimentation (Line 132-137), for replicability of the work.

We agree that employing a t-test was inappropriate on our part, due to the use of correlated data. Since the assumptions of a t-test were violated, we have removed the significance indicators on the graph and have removed that statistical description in ‘Material and Methods’. However, based on the established low variance in sensor measurements and the noticeable difference between the measured respiration for the Undiluted River Water sample in the 0.9 L headspace experiment and all five other experiments, we can still confidently conclude that microbial activity is detectable and a larger headspace volume enhances measurement sensitivity, thus supporting the methodology developed subsequently.

Reviewer:

L380 “Headspace CO2 concentration had a linear correlation with the 380 dilution of the river water,” between 25 and 100% strength (Fig 4; R2 = 0.986) and all other data with R2 in Fig 4.

AND

L287 “This technique confirmed the metabolic profile with tight correlation to CO2 data (R2=0.99; Fig 4). 

The R2 values reported are much higher than would be expected from the scatter seen in the data in the figures. The only explanation I can think of is that the data were averaged before calculating R2, which is the difference between the two plots shown here: https://i.imgur.com/GDZmrjv.png . This artificially deflates variance by performing stats on already processed data.

Authors:

The reviewer is correct, and the statistical values were calculated based on averages. In an attempt to fit all of the graphs into an aesthetically streamlined format, presenting multiple assessments per quadrant, the statistical analyses were incorrectly collapsed into a format that, as the reviewer points out, deflates variance. 

We have expanded them all as recommended by the reviewer and avoided passing the correlations through zero. We kept them on separate graphs for ease of audience interpretation, as single images become too busy. (Lines 350-363). We apologize for the conflation of the statistical values, and are grateful to the reviewer for encouraging the more rigorous analysis and stringent interpretation.

Reviewer:

L414 Fig 5 caption doesn’t match the figure. I don t understand what Fig 5 is showing.

Authors:

Thank you for bringing the lack of articulation to our attention. We’ve expanded the figure caption to provide a clearer description of the figure. We hope this brings clarity: if it is still unclear, we are happy to edit further. (Line 396-401).

---

## [Decision Letter · Decision Letter 1]

20 Jan 2021

PONE-D-20-04971R1

Canary in the coliform mine: Exploring the industrial application limits of a microbial respiration alarm system

PLOS ONE

Dear Dr. Stone,

Thank you for submitting your manuscript to PLOS ONE. After careful consideration, we feel that it has merit but does not fully meet PLOS ONE’s publication criteria as it currently stands. Therefore, we invite you to submit a revised version of the manuscript that addresses the points raised during the review process.

We look forward to receiving your revised manuscript.

Kind regards,

Dawei Zhang, Ph.D.

Academic Editor

PLOS ONE

Reviewers' comments:

Reviewer's Responses to Questions

**Comments to the Author**

1. If the authors have adequately addressed your comments raised in a previous round of review and you feel that this manuscript is now acceptable for publication, you may indicate that here to bypass the “Comments to the Author” section, enter your conflict of interest statement in the “Confidential to Editor” section, and submit your "Accept" recommendation.

Reviewer #2: (No Response)

Reviewer #3: (No Response)

2. Is the manuscript technically sound, and do the data support the conclusions?

Reviewer #2: Partly

Reviewer #3: Partly

3. Has the statistical analysis been performed appropriately and rigorously? 

Reviewer #2: No

Reviewer #3: Yes

4. Have the authors made all data underlying the findings in their manuscript fully available?

Reviewer #2: Yes

Reviewer #3: Yes

5. Is the manuscript presented in an intelligible fashion and written in standard English?

Reviewer #2: Yes

Reviewer #3: Yes

6. Review Comments to the Author

Reviewer #2: The authors present an interesting method for monitoring respiration by quantifying CO2. The device is tested in proof of concept experiments focusing on detection of changes to baseline respiration rates in each system – river water and activate sludge. Overall the data interpretation and modeling need to be improved.

Data and approach – It is unclear why the lumped mass transfer coefficient was fit. This could be evaluated using a control experiment. The potential issue in simultaneously fitting the lumped mass transfer coefficient and the CO2 production rate is that these parameters are likely to be correlated in ways that suggest potential for non-unique optimization/fit. The manuscript describes using R2 for assessing the two parameter fit (r and KLA). It would be helpful to establish that a one parameter model is not capable of describing the data, and then using an information criteria metric (e.g., Akaike) to establish the additional fitting parameter is warranted. A superior approach would be to independently determine KLA (fit to a control test) so as to only fit the rate parameter (i.e., what the device aims to measure) to the experimental data.

What is the implication of the variability in ambient CO2? The manuscript (Line 284) suggests that the baseline is 385 +/-45 ppm. What is the +/- here? The manuscript implies on the next line that it may be a standard deviation, but it is unclear. Since the monitoring device is based on differencing with this ambient condition (Line 127-128), what is the implication on the sensitivity of the motorizing? Later in the manuscript (Line 380-390) there is discussion about detection limit in terms of CFUs. How does the CFU detection limit depend on the substrate and nutrient condition of the water? That is, is using a detection limit in terms of CFUs system dependent? It stands to reason this type of conversion of the detection limit would depend on the rate at which the microbial community can oxidize the carbon. Seemingly this would depend on the form of the carbon present, the type of microbial community, and any limitations on carbon utilization (e.g., carbon, nutrient, and inhibitor/competitor concentrations).

Application – What is unclear from the work is the extent to which the CO2 monitoring system can help in application. The data show that the microbial community respond to perturbations established by adjusting chlorine concentration and pH, but what about monitoring a real plant? How much variability is present in the CO2 concentrations, and what is the time-scale of this variability compared to the time-scale of an upset in the community? Would upsets be detected earlier using this type of CO2 monitor? Also, how do the authors envision going from a beaker type test to a large, open top aeration tank? Is this a sample and test method, or is there potential for real time monitoring of the process? If the latter, is there a need for an array of sensors, or some sort of gas collection device? I raise these questions because the manuscript implies that the method can help in industrial settings without actually testing industrial settings.

River water vs activated sludge – the manuscript is often redundant in terms of the comparison between the two waters. The rationale for investigating these two end members of microbial activity are clear and can be stated once – ideally in the methods section. As I understand the manuscript, the results using the river water were not meaningful (Lines 290-298 suggest confounding factors limited the sensor or its interpretation). Unfortunately, this makes inclusion of the river water concept and data substantially less important. I recommend that the river water aspect be removed from the manuscript, or repeated using an experimental approach which resolves the issues related to the apparatus.

Minor items

Figure 3a – the use of predicted here for the modeled CO2 is incorrect. The manuscript states that the modeling results shown in figure 3a are a fit. Predictions have no fitted parameters (i.e., parameters are determined independent of the experiment being modeled). The modeling results for the 0.25 L case should be smooth though there appears to be a small increase then decrease in modeled CO2 around 2 hr. If this feature is real (i.e., the plot is small and the data line may be misleading the eye), then what in the model can account for the presence of this type of behavior? See above concern about the two parameter fit.

Reviewer #3: This manuscript provides a methodology to detect aberrations in microbial growth in an industrial setting. The advantage of CEMS is its utilization of sustainable, existing technology to function as a first-pass ‘canary’. Overall the manuscript is technically sound after the first revision. However, we would like to see more details on the points that we made below.

1. For figure 3a & b:

For Figure 3a, there is a discontinuation of the undiluted 0.90 L headspace experimental data between 3 - 6 hr point. We would like to see more explanation on how the measurement was done in specific.

2. For the description of figure 3 in the main manuscript, refer to the mathematical appendix for the equations and parameters to more clearly explain the regression.

3. L 416-421: The authors stated that this work demonstrates an early warning system for such significant contamination events (20 CFU/ml to 10300 CFU/ml). It would make a stronger argument if the authors explain what other methods are there for this kind of detection and give a comparison why this system would be better than the others.

7. PLOS authors have the option to publish the peer review history of their article (what does this mean?). If published, this will include your full peer review and any attached files.

Reviewer #2: No

Reviewer #3: No

---

## [Author Response · Author response to Decision Letter 1]

28 Jan 2021

Response to the generous editorial and reviewer comments have been provided as a separate PDF document (Response to Reviewers) during the submission process, as recommended throughout the editorial process. It allows for the formatting necessary for a coherent response, especially in the modelling sections.

---

## [Decision Letter · Decision Letter 2]

17 Feb 2021

Canary in the coliform mine: Exploring the industrial application limits of a microbial respiration alarm system

PONE-D-20-04971R2

Dear Dr. Stone,

We’re pleased to inform you that your manuscript has been judged scientifically suitable for publication and will be formally accepted for publication once it meets all outstanding technical requirements.

Kind regards,

Dawei Zhang, Ph.D.

Academic Editor

PLOS ONE

Additional Editor Comments (optional):

Reviewers' comments:

Reviewer's Responses to Questions

**Comments to the Author**

1. If the authors have adequately addressed your comments raised in a previous round of review and you feel that this manuscript is now acceptable for publication, you may indicate that here to bypass the “Comments to the Author” section, enter your conflict of interest statement in the “Confidential to Editor” section, and submit your "Accept" recommendation.

Reviewer #1: All comments have been addressed

Reviewer #3: All comments have been addressed

2. Is the manuscript technically sound, and do the data support the conclusions?

Reviewer #1: Yes

Reviewer #3: Yes

3. Has the statistical analysis been performed appropriately and rigorously? 

Reviewer #1: Yes

Reviewer #3: Yes

4. Have the authors made all data underlying the findings in their manuscript fully available?

Reviewer #1: No

Reviewer #3: Yes

5. Is the manuscript presented in an intelligible fashion and written in standard English?

Reviewer #1: Yes

Reviewer #3: Yes

6. Review Comments to the Author

Reviewer #1: The paper has been extensively modified from its first submision and in my opinion the reviewer comments have been adequately addressed. I asked for a minor revision so that the authors can ensure that the journals data policy has been completely fulfilled (it may be necessary to share raw data via a repository).

Reviewer #3: (No Response)

7. PLOS authors have the option to publish the peer review history of their article (what does this mean?). If published, this will include your full peer review and any attached files.

Reviewer #1: No

Reviewer #3: No

---

## [Editor Report · Acceptance letter]

23 Feb 2021

PONE-D-20-04971R2 

Canary in the coliform mine: Exploring the industrial application limits of a microbial respiration alarm system 

Dear Dr. Stone:

I'm pleased to inform you that your manuscript has been deemed suitable for publication in PLOS ONE. Congratulations! Your manuscript is now with our production department. 

Kind regards, 

on behalf of

Dr. Dawei Zhang 

Academic Editor

PLOS ONE